# Water and Nitrogen Coupling on the Regulation of Soil Nutrient–Microbial Biomass Balance and Its Effect on the Yield of Wolfberry (*Lycium barbarum* L.)

**DOI:** 10.3390/plants12152768

**Published:** 2023-07-25

**Authors:** Juan Yin, Zhenghu Ma, Yingpan Yang, Bin Du, Fubin Sun, Zhen Yang

**Affiliations:** 1School of Civil and Water Engineering, Ningxia University, Yinchuan 750021, China; 12020140084@stu.nxu.edu.cn (Z.M.); 12021140105@stu.nxu.edu.cn (Y.Y.); 12022140121@stu.nxu.edu.cn (B.D.); 12022131147@stu.nxu.edu.cn (F.S.); 12021130931@stu.nxu.edu.cn (Z.Y.); 2Ministry of Education Engineering Research Center for Modern Agricultural Water Resources Efficient Utilization in Dry Areas, Ningxia University, Yinchuan 750021, China; 3Ningxia Water-Saving Irrigation and Water Resources Control Engineering Technology Research Center, Ningxia University, Yinchuan 750021, China; 4The Scientific Research Institute of the Water Conservancy of Ningxia, Yinchuan 750021, China

**Keywords:** water and nitrogen regulation, soil nutrients, soil microbial biomass, ecological stoichiometry, yield of wolfberries, TOPSIS model, agricultural ecosystem

## Abstract

Due to the problems of relatively fragile stability, the quality of soil in the drip-irrigated agricultural ecosystem has high spatial heterogeneity and experiences significant degradation. We conducted a two-year field plot study (2021–2022) in a typical region of the arid zone with the “wolfberry” crop as the research object, with three irrigation and three nitrogen application levels, and the local conventional management as the control (CK). Soil quality under experimental conditioning was comprehensively evaluated based on Principal Component Analysis (PCA) and Technique for Order Preference by Similarity to an Ideal Solution (TOPSIS), and regression analyses were carried out between the soil quality evaluation results and wolfberry yield. The results showed that short-term water and nitrogen regulation enhanced the soil nutrient content in the root zone of wolfberry to some extent, but it did not significantly affect soil carbon:soil nitrogen (C_soil_:N_soil_), soil carbon:soil phosphorus (C_soil_:P_soil_), and soil nitrogen:soil phosphorus (N_soil_:P_soil_). When the irrigation quota was increased from I_1_ to I_2_, the soil microbial biomass carbon, nitrogen, and phosphorus (C_mic_, N_mic_, and P_mic_) tended to increase with the increase in N application, but the microbial biomass carbon:nitrogen (C_mic_:N_mic_), microbial biomass carbon:phosphorus (C_mic_:P_mic_), and microbial biomass nitrogen:phosphorus (N_mic_:P_mic_) did not change significantly. The comprehensive evaluation of the principal components and TOPSIS showed that the combined soil nutrient–microbial biomass and its ecological stoichiometry characteristics were better under the coupled treatments of I_2_, I_3_, N_2_, and N_3_, and the overall soil quality under these treatment conditions was significantly better than that under the CK treatment. Under I_1_ irrigation, nitrogen application significantly increased the yield of wolfberry, while under I_2_ and I_3_ irrigation, the wolfberry yield showed a parabolic trend with the increase in nitrogen application. The highest yield was recorded in the I_2_N_2_ treatment in the first and second years, with yields of 9967 kg hm^−2^ and 10,604 kg hm^−2^, respectively. The coefficient of determination (explained quantity) of the soil quality based on soil nutrient–microbial biomass and the characteristics of its ecological stoichiometry for wolfberry yield ranged from 0.295 to 0.573. These findings indicated a limited positive effect of these indicators of soil on wolfberry yield. The short-term water and nitrogen regulation partly influenced the soil and soil microbial biomass in agroecosystems, but the effect on elemental balance was not significant. Our findings might provide theoretical support for managing the health of agricultural ecosystems.

## 1. Introduction

Ecological stoichiometry is the study of energy balance and the balance between elements, such as carbon, nitrogen, and phosphorus, maintenance of whole ecosystems [1]. The concept was introduced by Elser et al. A study published by Sterner and Elser on ecological stoichiometry in Ecology and Oikos marked the emergence of ecological stoichiometry as a systematic and mature discipline [2]. Ecological stoichiometry is widely used for studying the dynamic balance of nutrients in the ecosystem, microbial nutrient transformation, and biogeochemical cycling in terrestrial ecosystems [3,4,5]. The elements C, N, and P are important components of soil as they are responsible for soil nutrient balance and cyclic transformation. Improper and extensive land use and anthropogenic activities can significantly disrupt the dynamic balance of C, N, and P and their distribution characteristics in the soil [6]. Microbial biomass is considered to be a source and sink in the soil nutrient cycling process, and its level reflects the soil organic matter turnover rate, soil vigor, and the level of ecosystem productivity [7,8]. The balance of microbial biomass is related to its activity and also influenced by processes such as nutrient release from soil organic matter [9,10].

Compared with other ecosystems, agricultural ecosystems have various applications, and human interference plays a dominant role in the development and evolution of the whole ecosystem. The study of the assignment of nutrient elements, such as C, N, and P, in crops and soils and their stoichiometric characteristics is important to elucidate the nutrient acquisition pathways of agricultural soils, reveal the cycling and transport of nutrients, and the evolution of the characteristics of the soil [11]. At present, most studies on the ecological stoichiometry of agricultural ecosystems have mainly focused on the effects of fertilization and tillage on the characteristics of the ecological stoichiometry of C, N, and P in the soil [12,13] and on the effects of fertilization and crop species on the ecological stoichiometry of crops [14]. The soil microbial biomass level was rarely addressed, and the studies were all on annual crops. Perennial crops have well-developed root systems and consume a large amount of soil nutrients, which can decrease soil quality and cause imbalances in elements. Understanding the effects of anthropogenic activities on soil nutrients and soil microbial ecological stoichiometry characteristics in the root zone of perennial crops is important for determining the health of the soil in agricultural ecosystems.

Wolfberry (*Lycium barbarum* L.) is a perennial deciduous shrub in the family Solanaceae [15]. It has significant beneficial effects on the kidney and liver, such as enhancing immunity, alleviating fever, clearing heat, and moistening the lungs [16,17]. Wolfberry requires a large amount of water and fertilizer, and to cope with regional water scarcity, drip irrigation technology is widely used in wolfberry cultivation [18]. Although water and fertilizer can be saved using this technique, they are mainly concentrated in the root zone of wolfberry, which decreases the stability of the soil and increases spatial heterogeneity, leading to changes in nutrient elements present in the soil not known.

Water and nitrogen fertilizer are the components most in demand during the cultivation of the wolfberry crop, and the application of water and nitrogen are also the most frequent and severely disruptive activity in cultivation. Proper water conditions promote plant root development, thus enhancing nitrogen uptake efficiency and conversion rate [19]. In contrast, inadequate water supply inhibits nitrogen fertilizer and crop growth. Too much will reduce the efficiency of water and fertilizer use, affecting crop yields and causing a large amount of soil nitrogen loss or leaching to the deep soil layer to cause soil and groundwater pollution [20]. Currently, Water and Nitrogen related researches mainly focus on the effects aspects such as crop growth, yield, quality, water and fertilizer use efficiency, and soil base nutrients [21,22]. The effects on the ecological stoichiometry characteristics of soil nutrient-microbial biomass have not been addressed. Thus, we conducted this study with the following objectives: (1) Discover the mechanism of short-term water and nitrogen regulation on the ecological stoichiometric balance of soil-microbial biomass; (2) Clarifying the amount of integrated soil quality contributing to crop yield based on soil nutrient allometry-microbial biomass ecological chemometrics. to construct a water and nitrogen management model with high efficiency of utilization of water and fertilizer resources, high yield and quality of wolfberry, and balanced soil nutrients.

## 2. Results

### 2.1. Characteristics of the Contents of C_soil_, N_soil_, and P_soil_ and the Changes in the Ecological Stoichiometry

The contents of C_soil_, N_soil_, and P_soil_ in the root zone of wolfberry in 2021 and 2022 showed an increasing trend with the increase in the application of N under the same irrigation quota (Figure 1a–c), while C_soil_:N_soil_, C_soil_:P_soil_, and N_soil_:P_soil_ did not change significantly (Figure 1d–f). The content of C_soil_ showed maximum and minimum values under both I_1_N_3_ and I_2_N_1_ treatments; the content of N_soil_ increased with an increase in the application of N, and the content of P_soil_ was relatively high under I_2_ irrigation. C_soil_:N_soil_ was higher under the I_1_N_1_ treatment, while C_soil_:P_soil_ and N_soil_:P_soil_ were similar among treatments. C_soil_:N_soil_ was not significantly affected by water and N regulation or their interaction, but C_soil_:P_soil_ and N_soil_:P_soil_ were significantly affected by irrigation (*p* < 0.05).

### 2.2. Characteristics of C_mic_, N_mic_, and P_mic_, and the Changes in the Ecological Stoichiometry

The contents of soil C_mic_, N_mic_, and P_mic_ in the root zone of wolfberry in 2021 and 2022 gradually increased with the increase in the level of N application at the same irrigation level (Figure 2a–c), and the increase was more prominent when the irrigation amount increased from I_1_ to I_2_ gradient, but C_mic_:N_mic_, C_mic_:P_mic_, and N_mic_:P_mic_ did not change significantly (Figure 2d–f). The soil C_mic_ content showed the highest and lowest values under the I_2_N_3_ and I_1_N_1_ treatments, respectively, and the soil N_mic_ content was significantly lower under the N_1_ treatment than under the N_2_ and N_3_ treatments in both years. Also, the values of C_mic_:N_mic_ and N_mic_:P_mic_ were significantly different under the I_1_N_1_ treatment compared to that under other treatments. Soil C_mic_, N_mic_, and P_mic_ were significantly affected by irrigation and N application. Additionally, C_mic_:N_mic_, C_mic_:P_mic_, and N_mic_:P_mic_ were affected to different degrees by irrigation, N application, and their interaction.

### 2.3. Correlation between Soil Nutrient–Microbial Biomass and the Ecological Stoichiometry Ratio

The results of Pearson’s correlation analysis showed some differences in the correlation between soil nutrient–microbial biomass and their ecological stoichiometry ratios in 2021 (Figure 3a) and 2022 (Figure 3b). The correlation between C_soil_ and C_soil_:P_soil_ was positive and significant (*p* < 0.05), the correlation between N_soil_ and P_soil_, N_soil_:P_soil_ and C_soil_:P_soil_ was positive and highly significant (*p* < 0.01), the correlation between C_soil_:N_soil_ and C_soil_:P_soil_ was positive and significant (*p* < 0.05), and the correlation between N_mic_:P_mic_ and N_mic_, C_mic_:P_mic_ was positive and highly significant (*p* < 0.01). The correlation between C_mic_:N_mic_ and N_mic_ was negative and highly significant (*p* < 0.01), and the correlation between C_mic_:N_mic_ and C_mic_:P_mic_ was negative and highly significant (*p* < 0.01).

To further examine the correlation between soil nutrients and microbial biomass, a linear fit was performed on the mean values of the data collected for two years (Figure 4), and the results showed that C_soil_ was significantly and positively correlated with C_mic_, N_mic_, and P_mic_ (Figure 4a–c). N_soil_ was significantly and positively correlated with C_mic_ and N_mic_ (Figure 4d,e), but the correlation with P_mic_ content was not significant (Figure 4f). None of the correlations between P_soil_ and C_mic_, N_mic_, and P_mic_ were significant (Figure 4g–i).

### 2.4. Effect of Water and Nitrogen Regulation on the Yield of Wolfberry

The effect of water and nitrogen regulation on the yield of wolfberry was significant (Figure 5). The highest yield in 2021 and 2022 and the mean values for the two years were recorded under the I_2_N_2_ treatment, with an increase in the yield by 6.68%, 9.08%, and 7.90% compared to that under the CK treatment and 20.23%, 28.75%, and 24.48% compared to that under the lowest yielding treatment (I_1_N_1_). The yield of wolfberry increased with the increase in nitrogen application under I_1_ irrigation, and the yield of wolfberry first increased and then slightly decreased with the increase in nitrogen application under I_2_ and I_3_ irrigation. These results indicated that excessive nitrogen application does not increase the yield of wolfberry.

### 2.5. Integrated Evaluation of Soil Nutrient–Microbial Biomass and the Effect on the Stoichiometric Characteristics

#### 2.5.1. Evaluation of Soil Nutrient–Microbial Biomass and the Stoichiometric Characteristics Based on PCA

The effects of water and nitrogen regulation on the stoichiometric characteristics of soil–microbial biomass were analyzed by PCA, and the principal components were extracted based on the principle of explained variance > 1 and cumulative contribution ≥ 85% (Figure 6a,b, and Table 1). The contribution of the first principal component was 42.6% and 58.6%, the contribution of the second principal component was 30.5% and 18.7%, and the contribution of the third principal component was 14.2% and 13.8% in 2021 and 2022, respectively. The cumulative contribution of the first three principal components was 87.3% and 91.1%, which reflected all the indicators, suggesting the feasibility of the PCA.

The composite scores of the three principal components extracted under different water and nitrogen regulation treatments were calculated (Table 2). A higher score indicated a better combination of soil nutrient-microbial biomass and its stoichiometric characteristics under that treatment. The combined effect of each treatment in 2021 was I_2_N_3_ > I_3_N_3_ > I_2_N_2_ > I_1_N_3_ > CK > I_3_N_2_ > I_1_N_2_ > I_3_N_1_ > I_2_N_1_ > I_1_N_1_. In 2022, it was I_3_N_2_ > I_3_N_3_ > I_2_N_3_ > I_2_N_2_ > I_1_N_3_ > CK > I_1_N_2_ > I_3_N_1_ > I_2_N_1_ > I_1_N_1_. The combined soil nutrient-microbial biomass and its stoichiometric characteristics were better under I_2_, I_3_, N_2_, and N_3_-coupled treatments. The combined scores of I_1_, I_2_, I_3_, and N_1_ coupled treatments were negative, and the soil quality was poor. These findings indicated that the effect of N application on the soil nutrient-microbial biomass and the stoichiometric characteristics of wolfberry cultivation sites was significant.

#### 2.5.2. Evaluation of Soil Nutrient–Microbial Biomass and the Stoichiometric Characteristics Based on the TOPSIS Method

To ensure that the evaluated results were accurate, the TOPSIS model was applied to further examine the soil–microbial biomass and stoichiometric characteristics (Table 3). A higher fit *C_i_* value with the optimal treatment indicated a better overall quality of the treatment. The results of the analysis were consistent with those of PCA, indicating that both evaluation methods were reliable, and the interannual differences might be attributed to frequent human disturbances and climatic and environmental conditions.

### 2.6. Effect of Soil Quality on the Yield of Wolfberry

To determine the influence of soil quality on the yield of wolfberry in the root area, linear regression analysis was conducted, and the relationship between soil quality and wolfberry yield was comprehensively evaluated (Figure 7a,b). The coefficients of determination (R^2^) of the integrated evaluation results of soil quality and yield fitting curves were 0.472 and 0.295 in 2021 and 0.573 and 0.412 in 2022, respectively, and the parameters were positively correlated. The results suggested a positive effect of soil nutrient total–microbial biomass and its stoichiometric ratio on the yield of wolfberry.

## 3. Discussion

### 3.1. The Response of C_soil_, N_soil_, and P_soil_ Contents and Their Stoichiometric Imbalance to the Water and Nitrogen Coupling Regulation

By conducting a water and nitrogen control experiment for two years, we found that the content of C_soil_, N_soil_, and P_soil_ in the root area of wolfberry increased with the increase in nitrogen application under certain irrigation conditions. Similar results of this finding have been obtained in studies related to agricultural, grassland, and wetland ecosystems [23,24,25]. Such patterns occurred because under suitable water conditions, the input of exogenous N fertilizer can directly enhance the effective soil N nutrients, and also because suitable water and N conditions promote the conversion and uptake of inorganic salts, which can indirectly increase the soil carbon and N stocks [24]. Phosphorus fertilizer was used in the experiment under the same exogenous input conditions, mainly due to differences in the endogenous release of soil phosphorus pools under different irrigation gradients [26]. Additionally, the use of exogenous N fertilizer improved the physicochemical properties of the soil to a certain extent, stimulating the decomposition of root zone litter, soil humus, and microbial residues, which increased the effect of returning these materials to the field besides meeting the nutrients required for crop growth, resulting in a gradual increase in the C_soil_, N_soil_, and P_soil_ content in the root zone of wolfberry [27].

We also found that N_soil_:P_soil_ gradually increased with the increase in N application, but the changes in C_soil_:N_soil_ and C_soil_:P_soil_ were not significant. Thus, we concluded that N addition could significantly increase soil N_soil_:P_soil_, but C_soil_:N_soil_ was generally relatively stable and not significantly affected by the addition of exogenous nutrients [28]. The C_soil_:N_soil_:P_soil_ ecological stoichiometry characteristics are mainly influenced by soil nutrient content, but some studies have reported an extremely strong positive correlation between soil TN and SOC and found the effects of water and nitrogen regulation on the contents of soil SOC and TN to be almost identical [25]. The N application conditions provided a large source of inorganic N to the soil, and suitable moisture conditions stimulated the decomposition of litter and microbial residues to replenish the carbon pool and increase the release of carbon sources [29]. Soil phosphorus was easily adsorbed with soil by the adsorption of soil iron and aluminum oxides, and it was not easily absorbed by crops, while the accumulation and consumption of TN and SOC, which are the main structural components of soil, were relatively fixed, leading to more stable C_soil_:N_soil_ and C_soil_:P_soil_ [30]. Exogenous phosphorus was quantitatively supplemented and limited the release of endogenous phosphorus caused by irrigation which might be the main reason for the gradual increase in N_soil_:P_soil_ with the increase in the application of N.

### 3.2. The Response of C_mic_, N_mic_, P_mic_, and Their Stoichiometric Imbalance to the Water and Nitrogen Coupling Regulation

The dynamics of soil microbial biomass can reflect microbial populations, abundance, and activity, and its magnitude is closely related to the physicochemical properties of the soil [31]. Both experiments showed that the soil C_mic_, N_mic_, and P_mic_ contents in the root zone of wolfberry increased with the increase in irrigation water and then leveled off or even decreased. These findings indicated that the soil microbial biomass under irrigated conditions was lower than that in rain-fed farmlands, and a slight increase in precipitation significantly contributed to the increase in soil microbial biomass in scrubland and grassland, while a significant decrease in precipitation resulted in a significant decrease in soil microbial biomass [32]. This occurred because suitable moisture conditions provided water necessary for biological activities and also maintained the permeability of the soil to some extent, providing suitable conditions for the survival of soil microorganisms. Also, better moisture conditions increased the transformation of nutrient transport between plants and soil and accelerated the rate of mineralization of soil organic matter [33]. Irrigation conditions might cause nutrient loss to some extent, and the increase in the frequency of soil flooding might lead the soil interstices to be filled with water, resulting in poor aeration and the inhibition of soil microbial activity [34]. Some studies have shown that N application can significantly increase C_mic_, N_mic_, and P_mic_ in woodland soils, mainly because the input of exogenous N increases the effective and total N content in the soil and promotes the decomposition, conversion, and utilization of soil organic matter by microorganisms [35]. Also, N addition decreases microbial dependence on N, which in turn exacerbates plant inter-root phosphorus and carbon limitation and induces microorganisms to secrete large quantities of carbon and phosphorus hydrolases to meet the nutrient uptake [36]. We also found that soil C_mic_, N_mic_, and P_mic_ in the root zone of wolfberry gradually increased with the increase in N application at the same irrigation level.

The C_mic_:N_mic_:P_mic_ ratio is an important indicator of nutrient limitation in the soil [37]. The global mean values of C_mic_:N_mic_, C_mic_:P_mic_, and N_mic_:P_mic_ for soils were found to be 42, 6, and 1, respectively [36]. The soil C_mic_:N_mic_, C_mic_:P_mic_, and N_mic_:P_mic_ in the root zone of wolfberry were around 11.10–24.04, 11.12–18.42, and 0.70–1.29, respectively, with short-term water and nitrogen regulation. The ratio of N_mic_:P_mic_ was lower than the global average ratio, indicating that the N_mic_ and P_mic_ contents of agricultural soils in our study area were relatively balanced, probably because of exogenous nitrogen supplementation [38]. By comparing the different water and nitrogen treatments, we found that the I_1_N_1_ treatment was significantly higher, indicating that some nitrogen limitation occurred under this treatment. The C_mic_:P_mic_ ratio for all treatments and N_mic_:P_mic_ ratio for some treatments were higher than the global soil, and no significant change in C_mic_:P_mic_ between treatments was detected, while N_mic_:P_mic_ showed an overall increasing trend with the increase in irrigation water, which indicated that increasing the quantity of irrigation water can improve the efficiency of the use of soil P to some extent [39]. The measures of soil microbial C, N, and P elements were mainly influenced by the soil organic matter content, and a higher abundance of effective N and fast-acting phosphorus in the soil indicated a lower microbial C, N, and P elemental ratio [37]. Humans play a key role in fertilizing and irrigating agricultural ecosystems, which in turn increase the microbial load and the activity of microorganisms and accelerate the rate of humus mineralization, leading to changes in the ecological stoichiometry of soil microorganisms.

### 3.3. The Response Mechanism of Wolfberry Yield to Soil Quality under Water and Nitrogen Coupling Regulation

Irrigation and the application of nitrogen fertilizer can supplement the water and nutrient elements required for crop growth and development, acting directly on the plant to promote crop growth and development and enhance crop yield [40]. They can also influence the soil environment, affecting nutrient cycling and elemental transport within the soil by regulating the soil carbon pool, which in turn can affect the release of elements in the soil and the characteristics of elemental balance, thus, indirectly affecting crop growth, development, and yield by changing the soil environment [41]. The irrigation and fertilizer application levels significantly affected crop growth, and the interactive effect of water and nutrients on crop growth and yield was significant [42]. The coefficient of determination (R^2^) of soil quality based on the total soil nutrient-microbial biomass and its chemometric ratio indicated positive effects on the yield of wolfberry under short-term water and nitrogen regulation in the range of 0.295–0.573. Nutrients are essential for crop growth and development; during crop growth, soil nutrients are consumed in large quantities. The application of nitrogen fertilizer additions makes up for the lack of satisfy the deficit in soil water and nutrients and helps in meeting the demands of the crops for soil water and nutrients, thus increasing crop yield [43]. When the amount of nitrogen applied exceeds the amount absorbed by the crop, the excess nitrogen is stored in the soil and increases soil fertility through accumulation. When crops have certain nutrient requirements or if the soil environment changes, the nitrogen and other nutrients in the soil are converted into inorganic salts that are easily absorbed and utilized by crops through microbial activation and mineralization, which improves crop yield [44].

Soil nutrients in this study only involved total soil–microbial biomass and the characteristics of its chemometric ratio. The positive effect of soil quality based on soil nutrient total–microbial biomass and its chemometric ratio on the yield of wolfberry was limited. It indicated that there are limitations in the soil nutrient evaluation system based on total soil nutrients and microbial biomass to reveal the effects of soil quality on crops. Soil fast-acting nutrients, enzyme activities, microbial diversity, and other sensitive indicators are more obviously affected by the external environment. Therefore, it is necessary to reveal the characteristics of soil fast-acting nutrients, enzyme activities, and microorganisms and their effects on crop physiology, growth, yield, and quality under water and nitrogen regulation in subsequent studies. At the same time, this study regarded the major area of root distribution of drip–irrigated wolfberry (0–40 cm) as a whole and did not reveal in detail characteristics of the soil elemental balance at different depths under water and nitrogen regulation and its relationship with crop yield in stratified layers, so further research is needed. Lastly, it will be significant for the sustainable development of agriculture if large-scale instrumentation can be utilized to achieve continuous monitoring of various indicators of soil nutrients, on the basis of which an intelligent decision-making management system can be established by using artificial intelligence technology.

## 4. Materials and Methods

### 4.1. Study Area

The study was conducted in Ningxia Concentric County, China (105°42′23.05″ E, 37°10′36.98″ N, and 1228 m, Figure 8). The area has a temperate continental semi-arid climate, with an average rainfall of 270 mm; approximately 70% of the annual rainfall occurs between July and September. The long-term average annual evaporation is about 2387 mm, which is higher than the average annual precipitation. The average annual temperature is around 8.6 °C, and the difference between the day and night temperatures is large. The average duration of sunshine is 3024 h, and the frost-free period is around 154 d. Soil properties in the test area were light loams in the range of 0–40 cm, with soil porosity of 46%, field water holding capacity of 13.4% to 15.1%, and other physicochemical properties are shown in Table 4. The groundwater of this region is between 20 and 25 m deep, and precipitation is minimal and concentrated. Thus, the reliable source of water for irrigation is the Yellow River. The precipitation and average daily temperature recorded throughout the fertility period in the two years of the study are shown in Figure 9. The varieties of wolfberry used in this study were 8a and 9a Ningqi No. 7. The plants were grown 0.75 m apart, and a line distance of 3 m was maintained. The trunk diameter of wolfberry trees is 30–45 mm, and they grow 80–110 cm in length at the start of their reproductive period. Wolfberry has a four-stage critical reproductive period, including the spring tip period (stage 1: late April to mid-May), the flowering period (stage 2: late May to mid-June), the fruit ripening period (stage 3: late June to mid-August), and the defoliation period (stage 4: late August to early September).

### 4.2. Experimental Design

The experiment was conducted from April to October 2021 and 2022. Two factors, i.e., the irrigation quota and nitrogen application, were manipulated in the experiment. Three irrigation levels, including low water I_1_ (2160 m^3^ ha^−1^), medium water I_2_ (2565 m^3^ ha^−1^), and high water I_3_ (2970 m^3^ ha^−1^) levels, and three N application rates, including low N_1_ (165 kg ha^−1^), medium N_2_ (225 kg ha^−1^), and high N_3_ (285 kg ha^−1^) rates, were used in the experiment following the drip irrigation planting technical protocols [45]. The information on the experimental design is presented in Table 5. The local conventional management standard was used as the control (CK) with 2970 m^3^ ha^−1^ irrigation quota, 330 kg ha^−1^ N, 90 kg ha^−1^ P, and 150 kg ha^−1^ K. Each plot had 10 trees with more than 3 m of separation between plots of wolfberry trees to prevent mutual influence. The experiment included nine treatments and one control treatment, with three replications for each treatment and 30 test plots in total. A 6 m buffer zone surrounding each test plot was maintained, and the total study area was 675 m^2^. Urea (46% N), calcium superphosphate (12% P_2_O_5_), and potassium sulfate (50% K_2_O) were used as test fertilizers. Above–ground, off–frame drip irrigation was performed, and the drip pipe was laid parallel to the rows. Water and nitrogen were provided simultaneously as follows: 20% and 15% (once) during the spring tip period, 30% and 25% (twice) during the flowering period, 40% and 50% (four times) during the fruit ripening period, and 10% (once) during both defoliation periods. All field management practices and levels, except for irrigation and fertilization, were identical to those of the surrounding area.

### 4.3. Soil Sample Collection

The soil samples were collected in the middle and late reproductive stages using a soil auger at approximately 10 cm of the roots from the side of the installed drip head of the wolfberry tree. The sampling depth was 0–40 cm, and soil samples were collected from three points in each plot and mixed thoroughly to obtain a fresh soil sample (about 1 kg). After removing gravel and animal and plant residues, the soil samples were divided into two parts using the quadrat method. One part was sealed in a sterile plastic bag, which was placed in a refrigerated box and transported back to the laboratory at the earliest for evaluating microbial biomass. The other part was dried under ambient conditions away from light; then, sand grains >2 mm were removed and sieved through 2 mm and 0.149 mm soil samples, respectively, to determine the physicochemical properties of the soil.

### 4.4. Measurement Indices and Methods

#### 4.4.1. Soil Index Measurements and Methods

Soil organic carbon (C_soil_) content was determined by the oxidation of potassium dichromate using an external heating method. The content of total nitrogen (N_soil_) was determined by the semi-micro Kjeldahl method—flow injector method. The content of total phosphorus (P_soil_) was determined by NaOH melting, molybdenum antimony anti-color development, and the UV spectrophotometric method.

Soil microbial biomass carbon (C_mic_), nitrogen (N_mic_), and phosphorus (P_mic_) indicators were determined by the chloroform fumigation leaching method [46]. Each sample of soil (fresh weight: 10 g) was fumigated with chloroform for 24 h at 25 °C in a dark box, and another sample of fresh soil (10 g) without chloroform fumigation was taken from each sample as a control. C_mic_ and N_mic_ were extracted with an aqueous solution of 0.5 mol/L K_2_SO_4_, P_mic_ was extracted with a solution of 0.5 mol/L NaHCO_3_, and the unfumigated control soil was extracted with 100 mL of 0.5 mol/L NaHCO_3_. The concentrations of carbon and nitrogen were measured using an organic carbon analyzer (TOC-VCPH, Shimadzu, Japan) and a continuous flow analyzer (AA3, SEAL Analytical, Germany), respectively. The phosphorus concentration was measured colorimetrically (the molybdenum blue colorimetric method) using a Shimadzu UV spectrophotometer (UV-2450, Shimadzu, Japan). C_mic_, N_mic_, and P_mic_ were derived using uniform conversion coefficients of 0.45, 0.54, and 0.40, respectively [47].

#### 4.4.2. Methods for Measuring the Yield of Wolfberry

Wolfberry yield was determined using the weighing method. During the fruit ripening period, all plots of ripe fruit were collected in batches. The fresh fruit yield of wolfberries was determined using an electronic scale (AX5202ZH, 0.01 g, Ohaus, Parsippany, NJ, USA).

### 4.5. Data Processing and Analysis

The data were organized and modeled using Microsoft Excel 2021. Analysis of variance, correlation analysis, and principal component analysis was performed using the SPSS 22.0 software. Significance tests were performed using the LSD method (*p* < 0.05). The Origin 21.0 software was used for correlation charting.

#### 4.5.1. Principal Component Analysis

Principal component analysis (PCA) is a method of dimensionality reduction analysis where variables are selected based on the amount of variance explained by each indicator and the cumulative contribution until the selected variables represent the information of the original indicator. Then, the combined score of each principal component is calculated [48].

(1)The formula for calculating each principal component score is as follows:


(1)
$$\{\begin{array}{c} F_{1}=Z_{11}X_{1}+Z_{12}X_{2}+\cdots +Z_{1j}X_{j}\\ F_{2}=Z_{21}X_{1}+Z_{22}X_{2}+\cdots +Z_{2j}X_{j}\\ \vdots \\ F_{n}=Z_{n1}X_{1}+Z_{n2}X_{2}+\cdots +Z_{nj}X_{j} \end{array}$$


(2)The data for the calculation of the combined score are as follows:


(2)
$$F=\left(b_{1}F_{1}+b_{2}F_{2}+\cdots +b_{n}F_{n}\right)/100$$


Here, *F* is the composite score; *b*_1_, *b*_2_, ⋯, and *b_n_* are the variance contribution rates; *F*_1_, *F*_2_, ⋯, *F_n_* are the scores of each principal component.

#### 4.5.2. TOPSIS Method

(1)A matrix was constructed based on the original evaluation parameters [49]. Assuming there were m evaluation objects and *n* evaluation indicators, the original data was expressed as the matrix
Χ = (${Χ}_{ij}$) *m* × *n*; where, $X_{ij}$ represents the original data of the *j*th indicator of the *i*th treatment. The metrics were normalized as follows:
(3)$$Z_{ij}=\frac{X_{ij}}{\sqrt{\sum_{j=1}^{n}{Χ}_{ij}^{2}}}$$
Here, *i* = (1, 2, …, *m*) and *j* = (1, 2, …, *n*). 

(2)The normalization matrix can be expressed as Ζ = ($Z_{ij}$) *m × n*. The optimal and inferior vectors composed of the maximum and minimum values of each column were determined using Equations (4) and (5), respectively.
(4)$$Z^{+}=(Z_{max1},\;Z_{max2},\;\ldots ,\;Z_{maxn})\;$$
(5)$$\;\;\;Z^{-}=(Z_{min1},\;Z_{min2},\;\ldots ,\;Z_{minn})\;$$

(3)The Euclidean distances (*D*_i_^+^ and *D*_i_^−^) were determined using Equations (6) and (7).
(6)$$\;\;\;D_{i}^{+}=\sqrt{\sum_{j=1}^{n}{{(Ζ}_{\max\;j}-{\;Ζ}_{\mathrm{ij}})}^{2}}$$
(7)$$\;D_{i}^{-}=\sqrt{\sum_{j=1}^{n}{{(Ζ}_{\max\;j}-{\;Ζ}_{\mathrm{ij}})}^{2}}$$

(4)The fit of the *i*th treatment to the optimal solution $C_{i}$ was determined using the equation.
(8)$$C_{i}=\frac{D_{i}^{-}}{\left(D_{i}^{+}+D_{i}^{-}\right)}$$

## 5. Conclusions

Short-term water and nitrogen coupling had some regulatory effects on soil nutrient–microbial biomass in the root zone of wolfberry, but soil nutrient and microbial biomass ratios did not change significantly, i.e., soil nutrient balances were not significantly disturbed by the short-term water and N regulation. The absence of anthropogenic nutrient supply in agricultural ecosystems is likely to lead to soil elemental imbalances, and appropriate water and fertilizer application can significantly enhance soil quality based on nutrient–microbial biomass and its stoichiometric characteristics. Suitable water and nitrogen conditions significantly enhanced the yield of wolfberry; the I_2_N_2_ treatment of two years of yield compared with CK increased by 6.68%, 9.08%, which was used as a reference for water and nitrogen management of wolfberry under drip irrigation conditions. A small positive effect of total soil nutrient-microbial biomass and the stoichiometric ratios was recorded on the yield of wolfberry. Our findings provided support for the internal stability of the soil in agricultural ecosystems and the green–efficient-sustainable development of agriculture.

## Figures and Tables

**Figure 1 plants-12-02768-f001:**
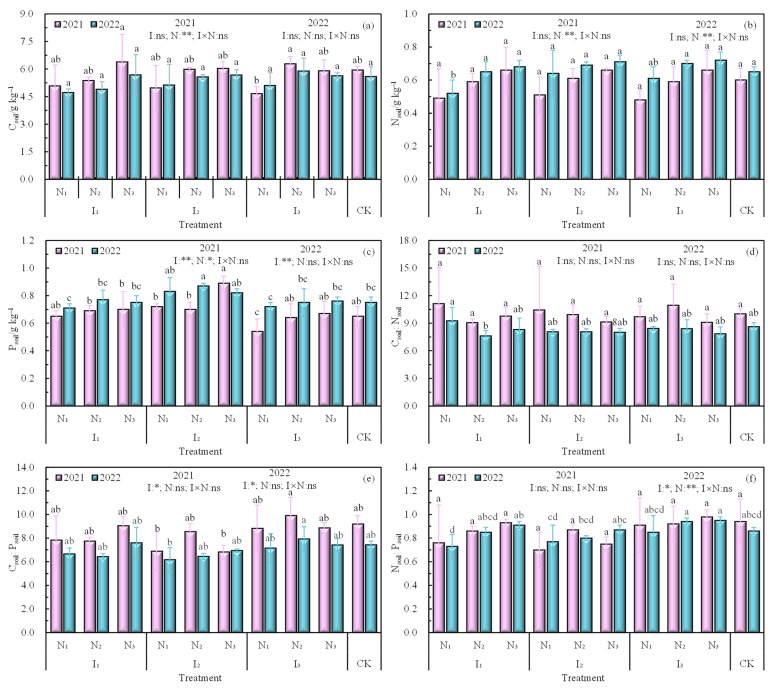
The contents of (**a**) C_soil_, (**b**) N_soil_, (**c**) P_soil_ and the ecological stoichiometric ratio of (**d**) C_soil_:N_soil_, (**e**) C_soil_:P_soil_, (**f**) N_soil_:P_soil_. Note: Different lowercase letters indicate significant differences between treatments (*p* < 0.05), ** indicates a highly significant level (*p* < 0.01), * indicates a significant level (*p* < 0.05), ns indicates no significant effect (*p* > 0.05), I is the irrigation quota, N is the amount of nitrogen applied, CK is the control.

**Figure 2 plants-12-02768-f002:**
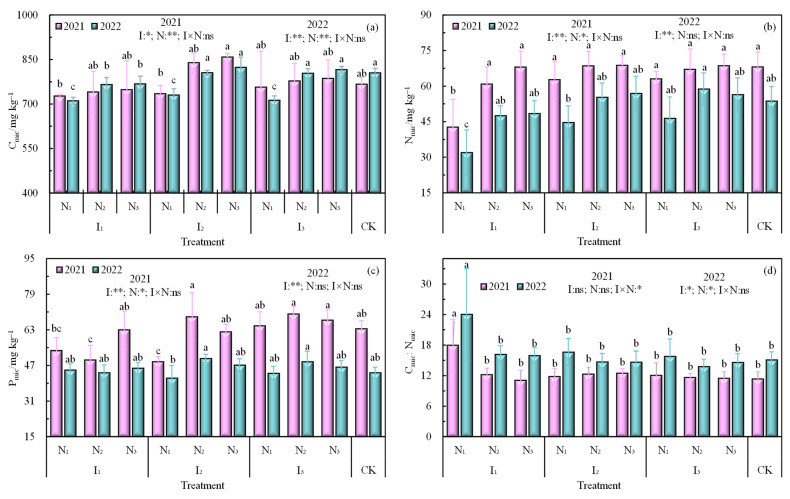
Characteristics The soil of (**a**) C_mic_, (**b**) N_mic_, (**c**) P_mic_ and the ecological stoichiometric ratio of (**d**) C_mic_:N_mic_, (**e**) C_mic_:P_mic_, (**f**) N_mic_:P_mic_. Note: Different lowercase letters indicate significant differences between treatments (*p* < 0.05), ** indicates a highly significant level (*p* < 0.01), * indicates a significant level (*p* < 0.05), ns indicates no significant effect (*p* > 0.05). I is the irrigation quota, N is the amount of nitrogen applied, CK is the control.

**Figure 3 plants-12-02768-f003:**
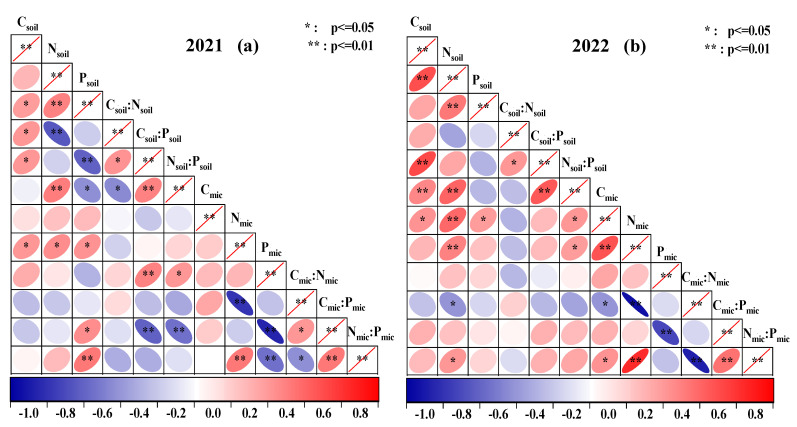
Correlation between soil nutrient–microbial biomass and the stoichiometric ratio.

**Figure 4 plants-12-02768-f004:**
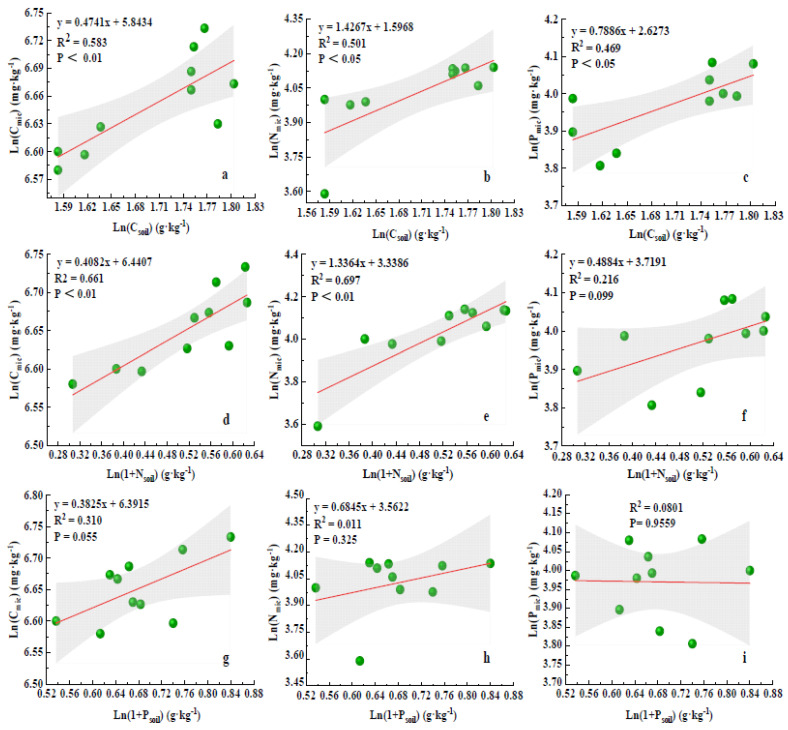
Correlation analysis of nutrient–microbial biomass. Note: (**a**) is the correlation between C_mic_ and C_soil_; (**b**) is the correlation between N_mic_ and C_soil_; (**c**) is the correlation between P_mic_ and C_soil_; (**d**) is the correlation between C_mic_ and N_soil_; (**e**) is the correlation between N_mic_ and N_soil_; (**f**) is the correlation between P_mic_ and N_soil_; (**g**) is the correlation between C_mic_ and P_soil_; (**h**) is the correlation between N_mic_ and P_soil_; (**i**) is the correlation between P_mic_ and P_soil_.

**Figure 5 plants-12-02768-f005:**
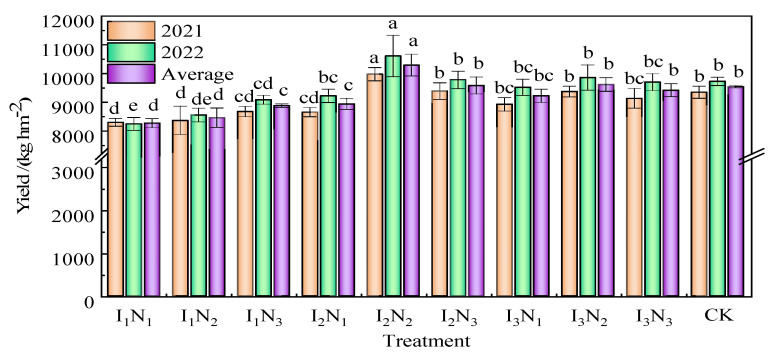
Effect of water and nitrogen regulation on the yield of wolfberry. Note: Different lowercase letters indicate significant differences between treatments (*p* < 0.05).

**Figure 6 plants-12-02768-f006:**
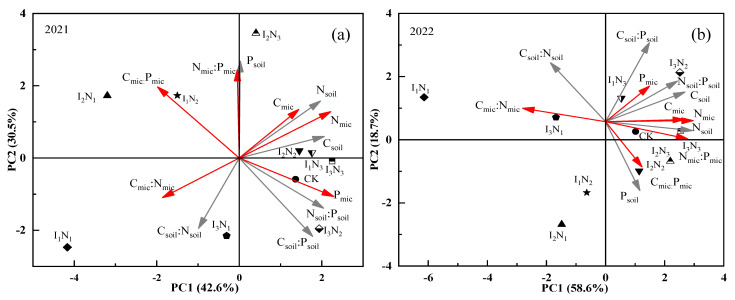
PCA of soil nutrient–microbial biomass and its stoichiometric ratio in (**a**) 2021 and (**b**) 2022. Black line: soil nutrients and the stoichiometric characteristics; red line: soil microbial biomass and the stoichiometric characteristics.

**Figure 7 plants-12-02768-f007:**
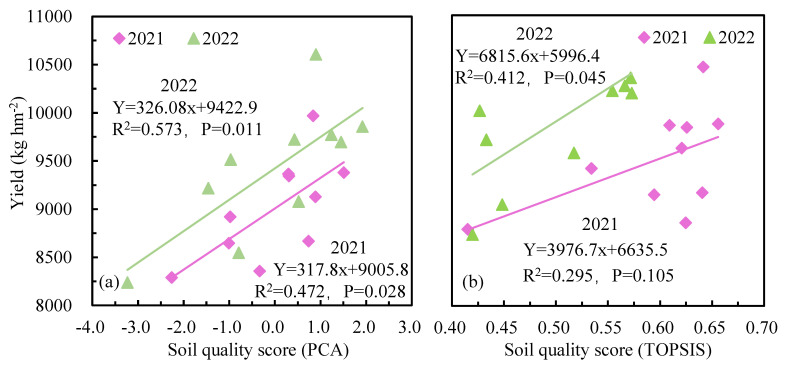
A linear regression relationship between soil quality and the yield of wolfberry, (**a**) Based on PCA, (**b**) Based on TOPSIS.

**Figure 8 plants-12-02768-f008:**
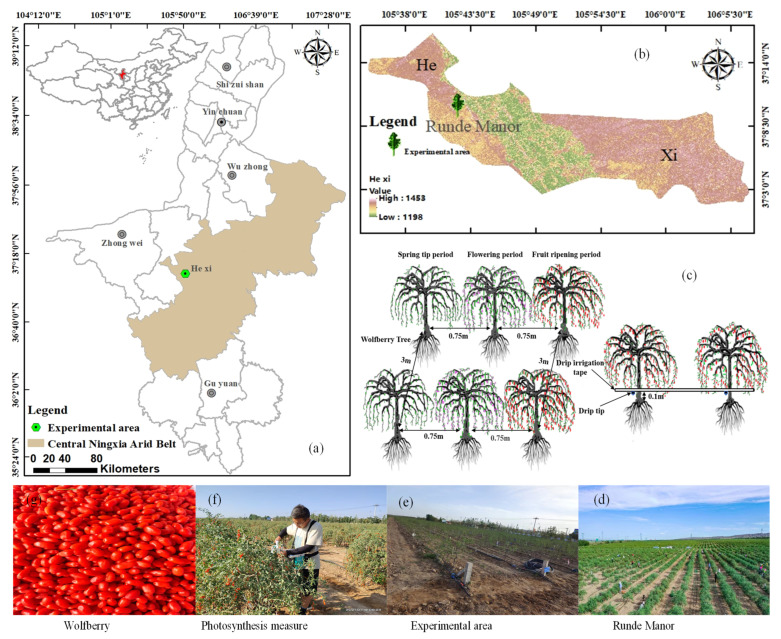
(**a**,**b**) The location of the study area on a map; (**c**–**g**) Schematic representation of the experimental arrangement.

**Figure 9 plants-12-02768-f009:**
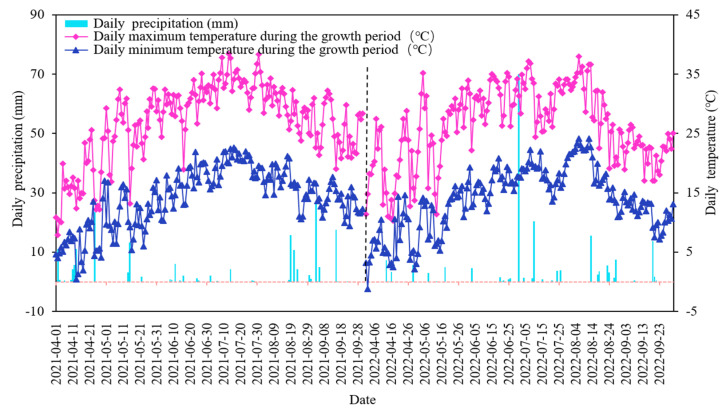
The daily precipitation and daily temperature recorded during the reproductive period of wolfberry in 2021 and 2022.

**Table 1 plants-12-02768-t001:** The results of PCA.

Year	Analysis Items	Principal Component 1	Principal Component 2	Principal Component 3	Analysis Items	Principal Component 1	Principal Component 2	Principal Component 3
2021	C_soil_	0.776	0.190	0.348	P_mic_	0.861	−0.339	0.273
	N_soil_	0.740	0.502	0.202	C_mic_:N_mic_	−0.701	−0.349	0.559
	P_soil_	0.009	0.854	0.487	C_mic_:P_mic_	−0.746	0.626	−0.096
	C_soil_:N_soil_	−0.371	−0.619	0.282	N_mic_:P_mic_	−0.018	0.776	−0.586
	C_soil_:P_soil_	0.669	−0.691	−0.157	Characteristics rate	5.11	3.66	1.71
	N_soil_:P_soil_	0.770	−0.442	−0.322	Contribution rate/%	42.6	30.5	14.2
	C_mic_	0.544	0.429	0.555	Cumulative contribution rate/%	42.6	73.1	87.3
	N_mic_	0.834	0.409	−0.233				
2022	C_soil_	0.891	0.307	0.125	P_mic_	0.493	0.369	0.772
	N_soil_	0.972	−0.093	0.096	C_mic_:N_mic_	−0.931	0.139	0.092
	P_soil_	0.385	−0.715	0.525	C_mic_:P_mic_	0.408	−0.469	−0.682
	C_soil_:N_soil_	−0.618	0.611	0.059	N_mic_:P_mic_	0.922	−0.178	−0.269
	C_soil_:P_soil_	0.492	0.818	−0.286	Characteristics rate	7.04	2.24	1.66
	N_soil_:P_soil_	0.806	0.421	−0.301	Contribution rate/%	58.6	18.7	13.8
	C_mic_	0.885	0.022	0.192	Cumulative contribution rate/%	58.6	77.3	91.1
	N_mic_	0.982	0.011	0.075				

**Table 2 plants-12-02768-t002:** A comprehensive evaluation of soil nutrient–microbial biomass and stoichiometric characteristics under water and nitrogen regulation.

Year	Treatment		Principal Component 1	Principal Component 2	Principal Component 3	Comprehensive Scores	Ranking	Year	Treatment		Principal Component 1	Principal Component 2	Principal Component 3	Comprehensive Scores	Ranking
2021	I_1_	N_1_	−4.17	−2.47	1.88	−2.26	10	2021	I_1_	N_1_	−6.14	1.35	0.83	−3.23	10
		N_2_	−1.50	1.73	−1.54	−0.33	7			N_2_	−0.64	−1.68	−0.75	−0.79	7
		N_3_	1.76	0.15	−0.40	0.74	4			N_3_	0.53	1.31	−0.26	0.52	5
	I_2_	N_1_	−3.20	1.72	−1.23	−1.01	9		I_2_	N_1_	−1.48	−2.68	−0.67	−1.46	9
		N_2_	1.45	0.20	1.11	0.84	3			N_2_	1.14	−0.99	3.00	0.90	4
		N_3_	0.40	3.46	1.98	1.51	1			N_3_	2.19	−0.68	0.61	1.24	3
	I_3_	N_1_	−0.31	−2.15	−1.39	−0.98	8		I_3_	N_1_	−1.68	0.71	−0.85	−0.97	8
		N_2_	1.94	−1.96	0.48	0.30	6			N_2_	2.51	2.13	0.35	1.92	1
		N_3_	2.25	−0.09	−0.27	0.89	2			N_3_	2.54	0.27	−0.67	1.45	2
	CK		1.36	−0.59	−0.63	0.31	5		CK		1.02	0.26	−1.57	0.43	6

**Table 3 plants-12-02768-t003:** A comprehensive evaluation of soil nutrient–microbial biomass and chemometric characteristics under water and nitrogen regulation, determined by the TOPSIS method.

Year	Treatment	Soil Nutrient–Microbial Biomass and Stoichiometric Characteristics Index												*D_i_^±^*	*D_i_^−^*	*C_i_*	Ranking
		C_soil_	N_soil_	P_soil_	C_soil_:N_soil_	C_soil_:P_soil_	N_soil_:P_soil_	C_mic_	N_mic_	P_mic_	C_mic_:N_mic_	C_mic_:P_mic_	N_mic_:P_mic_				
2021	I_1_N_1_	0.283	0.265	0.299	0.354	0.294	0.279	0.297	0.210	0.276	0.451	0.333	0.233	0.294	0.209	0.415	10
	I_1_N_2_	0.299	0.320	0.318	0.288	0.291	0.313	0.302	0.299	0.254	0.307	0.371	0.367	0.246	0.410	0.625	5
	I_1_N_3_	0.355	0.352	0.322	0.311	0.339	0.338	0.305	0.335	0.323	0.279	0.295	0.326	0.228	0.406	0.640	3
	I_2_N_1_	0.277	0.272	0.330	0.332	0.259	0.256	0.300	0.308	0.250	0.297	0.368	0.381	0.285	0.417	0.594	8
	I_2_N_2_	0.332	0.327	0.321	0.316	0.321	0.315	0.343	0.337	0.353	0.309	0.301	0.300	0.213	0.380	0.641	2
	I_2_N_3_	0.335	0.355	0.405	0.291	0.256	0.273	0.350	0.338	0.319	0.314	0.338	0.329	0.221	0.422	0.656	1
	I_3_N_1_	0.259	0.260	0.248	0.309	0.331	0.334	0.309	0.310	0.332	0.303	0.285	0.291	0.296	0.339	0.534	9
	I_3_N_2_	0.349	0.315	0.294	0.348	0.372	0.336	0.317	0.330	0.360	0.293	0.271	0.285	0.244	0.381	0.609	7
	I_3_N_3_	0.328	0.353	0.306	0.289	0.333	0.358	0.321	0.337	0.346	0.289	0.287	0.304	0.238	0.389	0.621	6
	CK	0.331	0.323	0.298	0.319	0.345	0.343	0.313	0.335	0.326	0.285	0.295	0.320	0.234	0.391	0.626	4
	*Z^±^*	0.355	0.355	0.405	0.354	0.372	0.358	0.350	0.338	0.360	0.451	0.371	0.381				
	*Z^−^*	0.259	0.260	0.248	0.288	0.256	0.256	0.297	0.210	0.250	0.279	0.271	0.233				
2022	I_1_N_1_	0.276	0.249	0.291	0.354	0.299	0.269	0.290	0.200	0.312	0.464	0.293	0.201	0.290	0.210	0.420	10
	I_1_N_2_	0.287	0.310	0.313	0.291	0.289	0.313	0.312	0.297	0.303	0.313	0.325	0.312	0.220	0.179	0.448	7
	I_1_N_3_	0.332	0.327	0.306	0.317	0.341	0.338	0.313	0.303	0.317	0.308	0.311	0.302	0.194	0.208	0.517	6
	I_2_N_1_	0.300	0.306	0.340	0.308	0.277	0.284	0.298	0.279	0.287	0.321	0.331	0.313	0.227	0.173	0.433	8
	I_2_N_2_	0.326	0.332	0.354	0.308	0.289	0.296	0.329	0.346	0.348	0.284	0.297	0.314	0.215	0.232	0.519	5
	I_2_N_3_	0.332	0.341	0.335	0.306	0.312	0.321	0.336	0.356	0.327	0.283	0.324	0.343	0.198	0.259	0.566	3
	I_3_N_1_	0.299	0.291	0.294	0.322	0.321	0.314	0.291	0.290	0.302	0.305	0.303	0.308	0.226	0.168	0.427	9
	I_3_N_2_	0.345	0.337	0.305	0.321	0.356	0.348	0.328	0.367	0.338	0.266	0.307	0.343	0.210	0.280	0.572	2
	I_3_N_3_	0.330	0.347	0.311	0.300	0.334	0.351	0.333	0.353	0.321	0.282	0.327	0.347	0.199	0.267	0.573	1
	CK	0.327	0.311	0.308	0.330	0.334	0.319	0.328	0.336	0.303	0.292	0.341	0.351	0.196	0.244	0.554	4
	*Z^±^*	0.345	0.347	0.354	0.354	0.356	0.351	0.336	0.367	0.348	0.464	0.341	0.351				
	*Z^−^*	0.276	0.249	0.291	0.291	0.277	0.269	0.290	0.200	0.287	0.266	0.293	0.201				

Note: *C_i_* indicates the fit degree, *Z^+^* indicates the ideal solution, *Z^−^* indicates the inverse ideal solution, *D_i_^+^* indicates the distance between each treatment and the ideal solution, and *D_i_^−^* indicates the distance between each treatment and the inverse ideal solution.

**Table 4 plants-12-02768-t004:** Physicochemical properties of the soil in the study area.

Year	Depth	pH	EC	Total N	Total P	Total K	Organic Material	Ammonium N	Nitrate N
	cm		μs cm^−1^	g kg^−1^				mg kg^−1^	
2021	0–20	8.00	1615	0.46	0.43	16.19	7.58	12.67	14.69
	20–40	8.20	1091	0.42	0.42	17.15	6.73	4.51	4.73
2022	0–20	8.06	1524	0.45	0.40	17.00	7.21	12.24	14.17
	20–40	8.12	963	0.41	0.41	17.00	6.28	4.42	4.82

**Table 5 plants-12-02768-t005:** The water and nitrogen management program of wolfberry.

Fertility Stages	2021							2022						
	Date	Irrigation Amount m^2^ hm^−2^			Nitrogen Amount kg hm^−2^			Date	Irrigation Amount m^2^ hm^−2^			Nitrogen Amount kg hm^−2^		
		I_1_	I_2_	I_3_	N_1_	N_2_	N_3_		I_1_	I_2_	I_3_	N_1_	N_2_	N_3_
Spring tip period	04–30	432	513.0	594	24.75	33.75	42.75	04–28	432	513.0	594	24.75	33.75	42.75
Flowering period	05–23	216	256.5	297	12.38	16.88	21.38	05–23	216	256.5	297	12.38	16.88	21.38
	06–18	432	513.0	594	28.88	39.38	49.88	06–15	432	513.0	594	28.88	39.38	49.88
Fruit ripening period	07–05	216	256.5	297	20.63	28.13	35.63	07–02	216	256.5	297	20.63	28.13	35.63
	07–15	216	256.5	297	20.63	28.13	35.63	07–15	216	256.5	297	20.63	28.13	35.63
	07–25	216	256.5	297	20.63	28.13	35.63	07–28	216	256.5	297	20.63	28.13	35.63
	08–05	216	256.5	297	20.63	28.13	35.63	08–08	216	256.5	297	20.63	28.13	35.63
Defoliation period	08–25	216	256.5	297	16.50	22.50	28.50	08–28	216	256.5	297	16.50	22.50	28.50
Total		2160	2565	2970	165	225	285		2160	2565	2970	165	225	285

## Data Availability

The data is contained within the article.

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
