# Peer review of "Water and Nitrogen Coupling on the Regulation of Soil Nutrient–Microbial Biomass Balance and Its Effect on the Yield of Wolfberry (Lycium barbarum L.)"

_plants, 2023, doi:10.3390/plants12152768_

Round 1
Reviewer 1 Report
Dear Authors,
I believe this paper is relevant to the readers of Plants. It presents a two-year study conducted in field conditions in an arid region with a precipitation level of 270 mm. Based on the provided pictures, it appears that the soil is sandy. It would be helpful if you could provide specific details regarding the soil texture and properties.
One of the challenges in this study is establishing a relationship between soil quality indices and crop yield. It is well known that crop yield is closely linked to the availability of nutrients in the soil. Given the conditions of plant cultivation, it is important to describe the concentrations of mineral N in the soil solution, as well as the levels of macro and micronutrients. While I understand that this was not the primary focus of your study, it would be beneficial to highlight the main attributes of soil fertility or the degree of soil fertility in the manuscript. By doing so, it would be possible to explore the relationship between crop yield and certain soil quality components, including microbiological parameters. To achieve this, I recommend sampling the soil from shallower layers, specifically at depths of 0-5 cm, 5-10 cm, 10-20 cm, and 20-40 cm in future trials. This approach would provide a better assessment of the total and available contents of carbon (C) and nitrogen (N), as well as other plant nutrients.
In summary, I agree that the findings described in the manuscript should be considered for publication. In the introduction, please provide a comprehensive review of soil quality indices, as well as their potential interactions with irrigation and the levels of added nitrogen in the soil. Additionally, it would be valuable to sample the soil solution in future studies to accurately evaluate the soil fertility index and its relationship with crop yield. While soil quality microbial or derived indices are relevant for soil preservation and environmental assessments, they are not directly correlated with crop yield. Rather, they only partially explain the crop indices discussed and determined, as demonstrated in this study. Therefore, in the discussion section, please address the potential limitations and flaws associated with using soil microbial quality indices to explain crop yield.
With the suggested changes, including improvements to the introduction, description of soil texture, soil mineral N availability (ammonium and nitrate), and a more thorough discussion, I believe this manuscript should be considered for further analysis and publication.
Sincerely,
The manuscritp was well written and the English grammar and style require only minor changes.
Author Response
Response to Reviewer 1 Comments
Dear reviewer:
Thank you for taking your time to review our article and give us your opinion. According to your comments, we have made a comprehensive revision of the article, and we would like to explain the revision of the article so that you can review it better, thank you again for your valuable comments!
Point 1: I believe this paper is relevant to the readers of Plants. It presents a two-year study conducted in field conditions in an arid region with a precipitation level of 270 mm. Based on the provided pictures, it appears that the soil is sandy. It would be helpful if you could provide specific details regarding the soil texture and properties.
Response 1: Thank you for your expert comments. According to your comments, we have added to the soil properties content of the study area. Based on our field sampling and analysis, the soil properties in the area were found to be light loams in the 0-40 cm range and sandy loams dominate below the 40 cm depth. We have added this content in the Materials and Methods Study Area section of the uploaded revised manuscript for re-review.
Point 2: One of the challenges in this study is establishing a relationship between soil quality indices and crop yield. It is well known that crop yield is closely linked to the availability of nutrients in the soil. Given the conditions of plant cultivation, it is important to describe the concentrations of mineral N in the soil solution, as well as the levels of macro and micronutrients. While I understand that this was not the primary focus of your study, it would be beneficial to highlight the main attributes of soil fertility or the degree of soil fertility in the manuscript. By doing so, it would be possible to explore the relationship between crop yield and certain soil quality components, including microbiological parameters. To achieve this, I recommend sampling the soil from shallower layers, specifically at depths of 0-5 cm, 5-10 cm, 10-20 cm, and 20-40 cm in future trials. This approach would provide a better assessment of the total and available contents of carbon (C) and nitrogen (N), as well as other plant nutrients.
Response 2: Thank you for your important comments. Your suggestion is very important for research, but our manuscript mainly discusses the total soil nutrients and microbial biomass, mainly to find out whether the short-term water and nitrogen regulation will have a significant effect on the soil and microbial biomass balance, and hence ignores the soil rapid-acting nutrient as well as the other major fertility indexes, which are actually more readily available for plants to take up and utilize, and which may be more obvious, and which will be the focus of our next research consideration. In fact, these indicators, which are more easily utilized by plants, may change more obviously, and this is also the focus of our next study.
The suggestion of split-level sampling has also been considered, and we are currently undertaking this work. In this manuscript, we only initially investigated the effect of water-nitrogen coupling on soil nutrients and microbial biomass in the 0-40 cm range of the root zone of wolfberry, because wolfberry, as a perennial crop, its root system is mainly distributed in the 0-40 cm range under drip irrigation environment. Your suggestions provide important guidance and help for our future research, thanks again!
Point 3: In summary, I agree that the findings described in the manuscript should be considered for publication. In the introduction, please provide a comprehensive review of soil quality indices, as well as their potential interactions with irrigation and the levels of added nitrogen in the soil. Additionally, it would be valuable to sample the soil solution in future studies to accurately evaluate the soil fertility index and its relationship with crop yield. While soil quality microbial or derived indices are relevant for soil preservation and environmental assessments, they are not directly correlated with crop yield. Rather, they only partially explain the crop indices discussed and determined, as demonstrated in this study. Therefore, in the discussion section, please address the potential limitations and flaws associated with using soil microbial quality indices to explain crop yield.
Response 3: We are grateful to the experts for their comments. Based on your comments, we have reviewed the literature in the relevant fields and have added the following content to the introduction of the article on the effects of irrigation and added nitrogen levels in soil on soil quality, as follows:" Proper water conditions promote plant root development, thus enhancing nitrogen uptake efficiency and conversion rate. In contrast, inadequate water supply inhib-its nitrogen fertilizer and crop growth, too much will reduce the efficiency of water and fertilizer use, affecting crop yields, and make a large amount of soil nitrogen loss or leaching to the deep soil layer to cause soil and groundwater pollution. Currently, Water and Nitrogen related researches mainly focus on the effects aspects such as crop growth, yield, quality, water and fertilizer use efficiency, and soil base nutri-ents." At the same time, we have made appropriate revisions to the rest of the article's introduction, which can be found in the red-lettered section of the introduction in the uploaded revised manuscript, which is available for review.
At the same time, based on your important comments and the opinions of other reviewers, we have made additions and modifications to the abstract, introduction, discussion, soil texture in Materials and Methods, and conclusions of the article, which are shown in the sections marked in red font in the uploaded revised manuscript. If there are any other problems with the article, please do not hesitate to let us know, and we will improve it again to enhance the quality of the article.
Point 4: Comments on the Quality of English Language:
The manuscritp was well written and the English grammar and style require only minor changes.
Response 4: Thank you for your comments. After the revision of the manuscript was completed, we commissioned a professional organization to invite native English-speaking experts to conduct a comprehensive check of the language presentation of the manuscript, in order to enhance the readability of the article and avoid unnecessary presentation errors and grammatical and logical problems. If you find that there are still major errors in language expression and logic during the review process, please let us know in time, and we will improve the article again to enhance the quality of the article.
Modification Tips
Dear reviewer, after revising and perfecting your comments, we have made additional adjustments to parts of the manuscript based on the comments of another reviewer, and all of them are highlighted in red font in the revised manuscript, please check them. If you still have questions about the manuscript and think this will affect the overall quality of the manuscript, please let us know in time, and we will improve it again. Thank you again.

Reviewer 2 Report
Dear author(s),
there are some inspiring insights thorough the manuscript and I tend to agree on its publication. However, there are few points that needs to be quickly addressed to improve its overall communication:
Title:
1/ clearly condensate the novelty and significance of the main discovery into a short and groundbreaking claim (significant shortening needed)
Abstract:
2/ strictly follow the established schema of writing academic Abstract: A/ introduction (urgency and significance of the research hypothesis); B/ principles of the methods used + key results; C/ conclusions (commercial and environmental impacts)
3/ do not use any abbreviations or technical terms in the Abstract , make it clear for anyone
4/ never report any results related to specific place or time, make sure your findings are globally applicable (higher level of generalization is needed)
5/ clearly indicate how will our international audience of readers benefit from these revelations (quantify the importance of your work)
Introduction:
6/ better address our global readership, make sure the topic is not limited to China
7/ remove all clusters of references to avoid reference overkill (prefer only 1 reference to support 1 claim)
8/ the complexity of phosphorus availability to organisms should be better explained, refer to Fig. 1 in paper "Novel sorbent shows promising financial results on P recovery from sludge water"
9/ make sure that this chapter fully introduces any reader into to the topic, explain all the terms, units, abbreviations, Latin and Greek letters, and the whole context that is necessary for anyone (including experts from other disciplines) to understand the following chapters
10/ the research hypothesis could be stated more clearly, condensate the research hypothesis into 1 short statement (or question) that will be subsequently confirmed or refuted, make sure the urgency and significance of the research hypothesis was justified in its environmental - economic nexus
Results:
11/ each Tab. and Fig. should be provided with caption that describes A/ what can be seen and B/ how is this relevant to the research hypothesis
12/ avoid data overkill, present only the most most industrially important results with a preference for those that are easier to interpret economically
Discussion:
13/ show more self-criticism to your work (do experiments on such a small volumes allow us to make any assumptions for commercial applications? are these samples representative? can all the methods and results be fully trusted? what are the weaknesses of the methods used? where do the main measurement inaccuracies arise? what are the limitations from a commercial point of view? are the lessons learned transferable to other fields?)
14/ he importance of process optimization and automation of managerial decisions using computer technology should be better explained (refer to papers "Sustainable Industry 4.0 Wireless Networks, Smart Factory Performance, and Cognitive Automation in Cyber-Physical System-based Manufacturing", "Sustainable Organizational Performance, Cyber-Physical Production Networks, and Deep Learning-assisted Smart Process Planning in Industry 4.0-based Manufacturing Systems" and "Artificial Intelligence Data-driven Internet of Things Systems, Real-Time Advanced Analytics, and Cyber-Physical Production Networks in Sustainable Smart Manufacturing")
15/ propose some improvements and direction for future research
16/ kindly note that C/N has been repeatedly and independently confirmed as an erroneous analysis in the last century because it does not reflect the availability of nutrients to living organisms (refer to paper "Advances in nutrient management make it possible to accelerate biogas production and thus improve the economy of food waste processing")
17/ compare your results in more depth with the existing literature, identify the main deviations and try to explain the mechanisms by which they may have been caused
Materials and Methods:
17/ the method must be presented in such a way that it can be reproduced anytime, by anyone, anywhere (do not create obstacles like referring to specific location etc.), please understand that the methodology must be described in a completely unambiguous way that does not allow for multiple interpretations (everyone who reads this chapter should get very precise instructions on how to repeat your procedure to achieve exactly the same results)
18/ each material/reactant and apparatus used needs to be presented in detail (serial number, setup, process parameters, manufacturer, country of origin, purity etc.)
Conclusions:
19/ do not repeat your methods and results again and again, please understand that the Conclusion chapter is not a summary of your work, present only original and industrially significant revelations that have the potential to expand the horizon of human knowledge (higher level of generalization is mandatory)
20/ clearly indicate whether the research hypotheses tends to be confirmed or not
Author Response
Response to Reviewer 2 Comments
Dear reviewer:
Thank you for taking your time to review our article and give us your opinion. According to your comments, we have made a comprehensive revision of the article, and we would like to explain the revision of the article so that you can review it better, thank you again for your valuable comments!
Title
Point 1: clearly condensate the novelty and significance of the main discovery into a short and groundbreaking claim (significant shortening needed)
Response 1: Thank you for your comments, which are particularly important. We investigated the effect of condition-specific " water and nitrogen coupled regulation" on the ecological stoichiometric balance of total soil nutrients and soil microbial biomass, and discussed the extent to which soil quality based on the indicators affects the crop yield. The study of nutrient balance in farmland was carried out by combining theories related to ecological environment. Due to the large number of experimental qualifications and concepts involved, it is difficult to compress the title of the article on a large scale in order to emphasize the characteristics of the study. We have simplified the title of the article as much as possible as follows:“Water and nitrogen coupling on the regulation of soil nutri-ent-microbial biomass balance and its effect on the yield of wolfberry (Lycium barbarum L.)”, please expert review it again. If you think it is still not to your satisfaction, please let us know, and we will do our best to optimize it in order to improve the article, thanks again!
Abstract
Point 2: strictly follow the established schema of writing academic Abstract: A/ introduction (urgency and significance of the research hypothesis); B/ principles of the methods used + key results; C/ conclusions (commercial and environmental impacts)
Response 2: Thank you to the experts for their input. Based on your comments, we have made detailed changes to the abstract section of the article. The details are as follows: Due to the problems of relatively fragile stability, the quality of soil in the drip-irrigated agricultural ecosystem has high spatial heterogeneity and experiences significant degradation. We conducted a two-year field plot study (2021-2022) in a typical region of the arid zone with the "wolfberry" crop as the research object, with three irrigation and three nitrogen application levels, and the local conventional management as the control (CK). Soil quality under experimental conditioning was comprehensively evaluated based on Principal Component Analysis (PCA) and Technique for Order Preference by Similarity to an Ideal Solution (TOPSIS), and regression analyses were carried out between the soil quality evaluation results and wolfberry yield. The results showed that short-term water and nitrogen regulation enhanced the soil nutrient content in the root zone of wolfberry to some extent, but it did not significantly affect soil carbon: soil nitrogen (Csoil: Nsoil), soil carbon: soil phosphorus (Csoil: Psoil), and soil nitrogen: soil phosphorus (Nsoil: Psoil). When the irrigation quota was increased from I1 to I2, the soil microbial biomass carbon, nitrogen, and phosphorus (Cmic, Nmic, and Pmic) tended to increase with the increase in N application, but the microbial biomass carbon: nitrogen (Cmic: Nmic), microbial biomass carbon: phosphorus (Cmic: Pmic), and microbial biomass nitrogen: phosphorus (Nmic: Pmic) did not change significantly. The comprehensive evaluation of the principal components and TOPSIS showed that the combined soil nutrient-microbial biomass and its ecological stoichiometry characteristics were better under the coupled treatments of I2, I3, N2, and N3, and the overall soil quality under these treatment conditions was significantly better than that under the CK treatment. Under I1 irrigation, nitrogen application significantly increased the yield of wolfberry, while under I2 and I3 irrigation, the wolfberry yield showed a parabolic trend with the increase in nitrogen application. The highest yield was recorded in the I2N2 treatment in the first and second years, with yields of 9,967 kg hm–2 and 10,604 kg hm–2, respectively. The coefficient of determination (explained quantity) of the soil quality based on soil nutrient-microbial biomass and the characteristics of its ecological stoichiometry for wolfberry yield ranged from 0.295 to 0.573. These findings indicated a limited positive effect of these indicators of soil on wolfberry yield. The short-term water and nitrogen regulation partly influenced the soil and soil microbial biomass in agroecosystems, but the effect on elemental balance was not significant. Our findings might provide theoretical support for managing the health of agricultural ecosystems.
(Note:Bolded parts are our modifications, unbolded parts show no modifications.)
Point 3: do not use any abbreviations or technical terms in the Abstract , make it clear for anyone
Response 3: Thank you, expert, for your important comments and corrections. Based on your suggestions, we have made detailed changes to the abstract section of the article, which are summarized in the response to question 2.
Point 4: never report any results related to specific place or time, make sure your findings are globally applicable (higher level of generalization is needed)
Response 4: Thank you for your important comments and corrections. It is our own limitation to ignore this research common sense, we according to your comments on the abstract related to the location of the locality of the present to be modified, but the time is mainly used to account for the time of our experiments, it is necessary to account for clearly, the details of the modification is shown in the reply to question 2, "Abstract section".
Point 5: clearly indicate how will our international audience of readers benefit from these revelations (quantify the importance of your work)
Response 5: Our research is focused on the important issues of soil nutrient spatial distribution heterogeneity and nutrient imbalance as well as soil degradation in agricultural ecosystems under drip irrigation conditions in arid and semi-arid regions. The study of the effects of agricultural activities of water and nitrogen regulation, the most important in agricultural production, on soil nutrient imbalance as well as soil quality, and hence crop yields, the study is useful and instructive for the health and sustainable development of drip-irrigated agroecosystems in arid and semi-arid regions of the global world.
Introduction:
Point 6: better address our global readership, make sure the topic is not limited to China
Response 6: Thank you for your expert comments. Your comments are significant to improve the quality of the article in general. In accordance with your comments, we have adjusted the order of the introductory part of the article and deleted or rewritten the geographically restricted descriptions to enhance the universality of the article's research content and application, and the details of which are shown in the introduction part of the uploaded revised manuscript.
Point 7: remove all clusters of references to avoid reference overkill (prefer only 1 reference to support 1 claim)
Response 7: Thank you for your expert comments. According to your comments, we have reorganized the references in the article. We have deleted all the references that support one point of view and retained only one, but we have retained all the references that support multiple points of view that require different documents. We believe this is essential and thank the experts again for their comments.
Point 8: the complexity of phosphorus availability to organisms should be better explained, refer to Fig. 1 in paper "Novel sorbent shows promising financial results on P recovery from sludge water"
Response 8: Dear Experts, Thank you for your comments. We are deeply aware that your comments are carefully considered, but our study mainly considered the characteristics of soil-microbial biomass ecological stoichiometric equilibrium influences in agricultural ecosystems. We mainly investigated the influence of the three elements of carbon, nitrogen, and phosphorus regulated by short-term water-nitrogen coupling, and the carbon to nitrogen ratio, carbon to phosphorus ratio, and nitrogen to phosphorus ratio were considered comprehensively in our study, and it was difficult for us to specifically analyze the complexity of the bio-phosphorus indicators and their relative importance in the limited space, which was not the focus of our study. And we believe that there are differences between phosphorus in agroecosystems and sludge, and that the relevant conclusions do not necessarily apply. Of course, your concerns are valid, and we may be able to validate them in subsequent studies where we are involved. Thank you again for your valuable comments.
Point 9: make sure that this chapter fully introduces any reader into to the topic, explain all the terms, units, abbreviations, Latin and Greek letters, and the whole context that is necessary for anyone (including experts from other disciplines) to understand the following chapters
Response 9: Thank you for the expert comments. Based on your comments, we have significantly revised and improved the introduction section of the article to ensure that it comprehensively introduces the reader to the topic of this study as well as the informational metrics and terminology necessary to understand the study. The details of these changes can be found in the red font in the Introduction section of the revised manuscript as submitted. Once again, we thank the experts for their detailed comments and guidance.
Point 10: the research hypothesis could be stated more clearly, condensate the research hypothesis into 1 short statement (or question) that will be subsequently confirmed or refuted, make sure the urgency and significance of the research hypothesis was justified in its environmental - economic nexus
Response 10: Thank you for your expert opinion. Based on your suggestions, we have rewritten and simplified the research hypothesis in the introduction of the article to make it more indirect and clear, and to better reveal the research topic of the article. Specifically rewritten as:" Thus, we conducted this study with the following objectives: (1) Discovering the mechanism of short term water and nitrogen regulation on the ecological stoichio-metric balance of soil-microbial biomass; (2) Clarifying the amount of integrated soil quality contributing to crop yield based on soil nutrient allometry-microbial biomass ecological chemometrics. "
Results:
Point 11: each Tab. and Fig. should be provided with caption that describes A/ what can be seen and B/ how is this relevant to the research hypothesis
Response 11: Thanks to the experts for the comments.Based on the experts' comments, we have further refined and confirmed the titles of the figures and tables appearing in the article, labeled the subfigures in the same subject with lowercase letters, and clarified the indexing of specific figures during the textual description and analysis of the article. For details, it is shown in the reddish colored parts of the font in the revised and uploaded manuscript Results and Analysis and Materials and Methods.
Point 12: avoid data overkill, present only the most most industrially important results with a preference for those that are easier to interpret economically
Response 12: Dear Reviewer, Thank you for your important comments on the article. In fact, for the foundation research and field experiment, our research monitoring data are not much, we monitored 6 indicators of soil carbon, nitrogen, phosphorus and microbial biomass carbon, nitrogen and phosphorus, and calculated 6 indicators of elemental stoichiometric ratio based on the measured data, totaling 12 indicators, which is a relatively normal or even small amount of data in field research. Each indicator of soil nutrient totals and microbial biomass are very important for the study of soil nutrient balance and spatial heterogeneity, and we did not streamline the data in order to make the presentation of the article more complete. The data in this article seem to be more complicated probably because we had more experimental treatments, and furthermore, in order to reduce the experimental errors, we conducted two consecutive years of field trials, which is very necessary to ensure the quality of the data in the field trials. We would like to ask the reviewers to double-check the data, and if there are any questions, please contact us in time, thank you again.
Discussion:
Point 13: show more self-criticism to your work (do experiments on such a small volumes allow us to make any assumptions for commercial applications? are these samples representative? can all the methods and results be fully trusted? what are the weaknesses of the methods used? where do the main measurement inaccuracies arise? what are the limitations from a commercial point of view? are the lessons learned transferable to other fields?)
Response 13: Dear Reviewer, Thank you for your comment. Based on your comments, we have revised and improved the discussion section of the manuscript. However, due to the limited number of studies in agricultural ecosystems and even fewer studies in perennial crops such as wolfberry, we had to refer to studies in forest and grassland ecosystems for further clarification, and some discussions may not have been presented perfectly.At the same time, the field experiment has more limitations of geographical environment, in order to reduce the error, we conducted two years of field experiment, which is the common method we use to reduce the experimental error at present. There are also errors in the measurement of samples, but we guarantee that errors are within control and do not affect the results. Regardless of the geographical environment and other differences, our test results are reliable for wolfberry or other perennial crops, but for annual crops and other areas, we still need a lot of experiments to verify these hypotheses. Specific revisions are shown in the uploaded revised manuscript discussion section in red font. We thank the reviewers again for their comments, and if you have any questions about this description and revisions that may affect the overall quality of the manuscript, please let us know, and we will spare no effort to improve it again.
Point 14: he importance of process optimization and automation of managerial decisions using computer technology should be better explained (refer to papers "Sustainable Industry 4.0 Wireless Networks, Smart Factory Performance, and Cognitive Automation in Cyber-Physical System-based Manufacturing", "Sustainable Organizational Performance, Cyber-Physical Production Networks, and Deep Learning-assisted Smart Process Planning in Industry 4.0-based Manufacturing Systems" and "Artificial Intelligence Data-driven Internet of Things Systems, Real-Time Advanced Analytics, and Cyber-Physical Production Networks in Sustainable Smart Manufacturing")
Response 14: Dear Expert, Thank you for your constructive comments. Your opinions are very important, the application of computer technology to realize artificial intelligence management and decision-making automation is a research tool that all researchers are trying to realize, which is significant for the improvement of work efficiency and cost control in various industries. At present, it is widely used in the efficient use of agricultural water and fertilizer resources, as well as intelligent decision-making irrigation, digital irrigation construction, etc., which is also the main direction of our team's further research. However, this manuscript was just a preliminary basic theoretical study, on which we hope to provide some experience accumulation and data support for the subsequent application of AI technology in this field. Our research idea is also to adopt this step-by-step approach, so as to better solve the scientific problem of soil nutrient balance ( healthy ) - green, efficient and sustainable development of agriculture. Once again, we thank you for your comments.
Point 15: propose some improvements and direction for future research
Response 15: Dear experts, in terms of the limitations of this manuscript, we believe that in future research, we should do more in-depth and detailed work at the theoretical level, such as stratification of the soil, monitoring more indicators of soil fertility, especially soil fast-acting nutrients, microorganisms, and enzyme activities and other sensitive indicators, to comprehensively establish the relationship between soil indicators and crop yield and quality, so that the dynamic equilibrium between soil nutrients and crops, and the water and fertilizer use efficiency can be maintained at the inter-annual scale. At the methodological level, as you mentioned in Q14, the importance of computer technology and automated management and decision-making systems is an important direction for the future development of this field. Continuous monitoring of soil nutrient indicators should be carried out with the help of large-scale instrumentation, and based on this, an intelligent decision-making management system should be established by utilizing artificial intelligence technology. In addition, it is necessary to reveal the influence mechanisms of different regional environments and agricultural activities on soil nutrient balances under different cropping patterns, so as to establish intelligent decision-making management systems for different crops according to local conditions.
Point 16: kindly note that C/N has been repeatedly and independently confirmed as an erroneous analysis in the last century because it does not reflect the availability of nutrients to living organisms (refer to paper "Advances in nutrient management make it possible to accelerate biogas production and thus improve the economy of food waste processing")
Response 16: Dear Experts, Thank you for your comments. Ecological stoichiometry is the study of energy balance and the balance between elements, such as carbon, nitrogen, and phosphorus, maintenance of whole ecosystems . Currently, the concepts of C/N, C/P and N/P are widely used in forest ecosystems, grassland ecosystems and wetland ecosystems health studies, and have been proved to be significantly affected by the number of years of grassland restoration and nutrient additions, etc. Studies have shown that they are closely related to nutrient uptake by forests and grasses and have important indications of the balance of nutrient uptake by plants as well as the health of soils. However, agricultural ecosystems are subject to great uncertainty due to frequent human interference and frequent crop turnover, so there are fewer related studies. Agriculture is the foundation of human civilization, and the health of agricultural ecosystems is closely related to people's lives. Improving the efficiency of agricultural water and fertilizer resources is only one aspect of the importance of agricultural ecosystems, but maintaining the balance of nutrients in agricultural soils is the key to achieving green and sustainable development. We believe that compared to the points you mentioned and the supporting research topics, agricultural ecosystems are more similar to forest and grassland ecosystems, and the related theories can be better verified by each other. Please let us know if you still have any concerns, and we will work on them again. Thank you again for your comments.
Point 17: compare your results in more depth with the existing literature, identify the main deviations and try to explain the mechanisms by which they may have been caused
Response 17: Dear Reviewer, Thank you for your comments. Based on your suggestions we have revised and improved the discussion section of the article and added the missing points in the article that you mentioned. The specific changes are shown in the red font in the discussion of the uploaded revised manuscript. If there are still some problems with the article, please let us know and we will again try our best to improve it in order to enhance the quality of the manuscript. Thanks again.
Materials and Methods
Point 18: the method must be presented in such a way that it can be reproduced anytime, by anyone, anywhere (do not create obstacles like referring to specific location etc.), please understand that the methodology must be described in a completely unambiguous way that does not allow for multiple interpretations (everyone who reads this chapter should get very precise instructions on how to repeat your procedure to achieve exactly the same results)
Response 18: Thanks to the expert for the important comments.Based on your comments, we have revised and improved the Materials and Methods section of the article, deleting the special locations and repetitive expressions, but retaining the expression " The study was conducted in Ningxia Concentric County, China " for our study area, which is necessary for the integrity of the experiment. This is necessary for the completeness of the experiment. Details are shown in the red section of the Materials and Methods section of the revised uploaded manuscript.
Point 19: each material/reactant and apparatus used needs to be presented in detail (serial number, setup, process parameters, manufacturer, country of origin, purity etc.)
Response 19: Thank you for your comments. Based on your comments, we have added information on the models, manufacturers, and places of origin of the instruments and equipment used in the tests, and the specific modifications are shown in the red-lettered section of Sample Tests and Methods in Materials and Methods in the revised and uploaded manuscript. Thank you again for your comments!
Conclusions:
Point 20: do not repeat your methods and results again and again, please understand that the Conclusion chapter is not a summary of your work, present only original and industrially significant revelations that have the potential to expand the horizon of human knowledge (higher level of generalization is mandatory)
Response 20: Thank you for your important comments , your suggestions are very important to improve the quality of the article. Based on your suggestions, we have reorganized and summarized the conclusion section of the article to make it more universal and not limited to the content of the article. The specific revision is " Short-term water and nitrogen coupling had some regulatory effects on soil nutrient-microbial biomass in the root zone of wolfberry, but soil nutrient and microbial biomass ratios did not change significantly, i.e., soil nutrient balances were not significantly disturbed by the short-term water and N regulation. The absence of anthropogenic nutrient supply in agricultural ecosystems is likely to lead to soil elemental imbalances, and appropriate water and fertilizer application can significantly enhance soil quality based on nutrient-microbial biomass and its stoichiometric characteristics. Suitable water and nitrogen conditions significantly enhanced the yield of wolfberry, the I2N2 treatment of two years of yield compared with CK increased by 6.68 %, 9.08 %, which was used as a reference for water and nitrogen management of wolfberry under drip irrigation conditions. A small positive effect of total soil nutrient-microbial biomass and the stoichiometric ratios was recorded on the yield of wolfberry. Our findings provided support for the internal stability of the soil in agricultural ecosystems and the green-efficient-sustainable development of agriculture."
Point 21: clearly indicate whether the research hypotheses tends to be confirmed or not
Response 21: Dear reviewer, The two hypotheses presented in the introduction are the main theme throughout our article, and we have finally made our hypotheses validated by analyzing and verifying the experimental data and discussing them in comparison with related studies.We conclude that " Short-term water-nitrogen coupling had some regulatory effects on soil nutrient-microbial biomass in the root zone of wolfberry, but soil nutrient and microbial biomass ratios did not change significantly, i.e., soil nutrient balances were not significantly disturbed by the short-term water and N regulation. A small positive effect of total soil nutrient-microbial biomass and the stoichiometric ratios was recorded on the yield of wolfberry."

Round 2
Reviewer 1 Report
Dear authors,
I have carefully read the revised manuscript, and I must say that all the issues highlighted in my report have been correctly addressed. More importantly, the authors have thoroughly recognized and described in the body of the manuscript all the actions necessary to understand the main drivers of yield for Wolfberry in future trials, as well as the key strategies and soil sampling scheme. The only aspect that deserves further improvement is the classification of the soil according to Soil Taxonomy or the FAO soil classification system.
With these comments in mind, I believe the paper should be accepted for publication in its current form.
Sincerely,